# Mucosal Immune Responses to Respiratory Syncytial Virus

**DOI:** 10.3390/cells11071153

**Published:** 2022-03-29

**Authors:** Megan V. C. Barnes, Peter J. M. Openshaw, Ryan S. Thwaites

**Affiliations:** National Heart and Lung Institute, Imperial College London, London W2 1NY, UK; m.barnes19@imperial.ac.uk (M.V.C.B.); p.openshaw@imperial.ac.uk (P.J.M.O.)

**Keywords:** RSV, bronchiolitis, infants, respiratory mucosa, Type-2 immunity

## Abstract

Despite over half a century of research, respiratory syncytial virus (RSV)-induced bronchiolitis remains a major cause of hospitalisation in infancy, while vaccines and specific therapies still await development. Our understanding of mucosal immune responses to RSV continues to evolve, but recent studies again highlight the role of Type-2 immune responses in RSV disease and hint at the possibility that it dampens Type-1 antiviral immunity. Other immunoregulatory pathways implicated in RSV disease highlight the importance of focussing on localised mucosal responses in the respiratory mucosa, as befits a virus that is essentially confined to the ciliated respiratory epithelium. In this review, we discuss studies of mucosal immune cell infiltration and production of inflammatory mediators in RSV bronchiolitis and relate these studies to observations from peripheral blood. We also discuss the advantages and limitations of studying the nasal mucosa in a disease that is most severe in the lower airway. A fresh focus on studies of RSV pathogenesis in the airway mucosa is set to revolutionise our understanding of this common and important infection.

## 1. Background

Respiratory syncytial virus (RSV) is a single-stranded, negative-sense RNA virus with a genome comprising 10 genes encoding 11 proteins [1]. While RSV triggers common cold symptoms in healthy adults, it poses a much greater threat to infants and the elderly [2,3,4]. The importance of understanding protective immunity is underlined by the fact that there is, as yet, no licenced vaccine for human use [5]. Even natural infection does not protect against reinfection, despite limited antigenic variability [6].

RSV causes the majority of cases of bronchiolitis in childhood and is the most common single cause of hospitalisation in children less than two years of age [7], with around 70% of hospitalisations occurring in the first six months of life [8]. Symptoms vary from mild or unapparent to severe respiratory distress and the requirement for mechanical ventilation. Although there are established risk factors for severe disease, the reasons for this diverse response are not fully understood. Risk factors include maternal smoking, social deprivation, pre-existing cardiovascular and/or lung disease [9], and pre-term birth (particularly in infants born before 35 weeks of gestation [10,11]). Given this burden of disease in infants born prematurely, neonatal immunology is particularly understudied. A review on the pre- and postnatal development of the immunological environment of the lung highlights the key mediators in innate and adaptive immunity in neonates (infants less than 28 days old) [12]. Even with these risk factors considered, term infants with no apparent underlying health conditions may still be susceptible [13], with risk factors for this cohort including sex, urban living [14], age, and attendance to day-care [15]. Genetic factors, for example specific single nucleotide polymorphisms (SNPs), have been investigated in the context of RSV disease susceptibility and severity in genome-wide association studies (GWAS) [16,17]. RSV is typically associated with greater symptom severity than other respiratory viruses [18] and associations with long-term respiratory disease have been widely studied. Early studies linked RSV bronchiolitis with the subsequent development of childhood wheeze and progression to asthma [19,20,21]. While more recent, large-scale, multicentre studies, such as COAST [22] and ALSPAC [23], have confirmed a causal link with wheeze (reviewed elsewhere [24]), a direct role in the development of asthma is less certain. This uncertainty may result from our limited understanding of the immune response to respiratory viruses in infants and how acute infection influences the temporal evolution of the immunological and microbial environment of the airway [25]. Although it is not the main focus of the review, the rapid immune and microbial changes in the respiratory tract following birth present a major challenge in understanding the response to RSV infection in infants and warrants further discussion. The recent discoveries of an IL-33-mediated, Type-2-skewed environment in the healthy neonatal murine lung [26], and the susceptibility of neonatal regulatory B cells (nBregs) to RSV infection [27], further emphasize the importance of studying this crucial developmental period. Furthermore, this challenge is compounded by the difficulty of sampling the site of disease in the upper, and particularly lower, respiratory tract. These difficulties have resulted in the development of animal models of RSV infection, which enable dissection of the mechanisms underpinning RSV susceptibility and disease severity.

Observations of a paradoxical decrease in upper airway viral load (VL) in severe RSV disease [28,29] and non-significant differences in VL across severity cohorts in another study [30] suggest that viral burden is not the primary factor driving severity, despite VL correlating with disease severity in mice [31]. However, longitudinal measurements of VL following disease onset indicates that hospitalised infants with more severe disease have slower viral clearance than those with moderate disease [29]. By contrast, elevated inflammatory responses in the airway have been associated with disease severity, suggesting that the immune response to infection contributes to disease pathogenesis [28,32,33,34,35,36,37,38,39]. There are several factors which need to be considered when studying the infant immune response to RSV infection. First, patients can be categorised into mild, moderate, or severe disease groups, with different studies utilising a variety of definitions and criteria, such as global respiratory severity score (GRSS) [40], clinical disease severity score (CDSS) [29], and hospital admission [28]. Greater standardisation of severity categorisation would aid the ability to interrelate studies. Secondly, environmental factors and the development of the immune system at the time of RSV exposure can influence susceptibility and severity, as recently reviewed elsewhere [41]. The legacy of these pre-infection differences confounds our ability to distinguish responses to infection from baseline factors that predispose infants to RSV disease. Finally, an important factor to consider is persistent elevation of immune mediators following infection, and whether specific cellular responses cause long-term damage to respiratory health. Specifically, the imbalanced secretion of Type-1, Type-2, and Type-17 cytokines by various immune cells has been investigated in the context of disease severity, both at the time of primary infection, and during reinfection later in life.

These Type-1, Type-2, and Type-17 responses are associated with the release of specific cytokines that are aligned with T-helper (Th)-1, Th-2, and Th-17, or cytotoxic Type-1 (Tc1), Tc2 and Tc17 responses [42,43,44]. Frequencies of ILC2s, which function in a similar manner to Th-2 cells, and concentrations of Type-2 cytokines were recently demonstrated to positively correlate with disease severity [30] in the nasopharyngeal aspirates (NPAs) of RSV-infected infants. Other studies have also observed increased numbers of Type-2 lymphocytes and their cytokines, which positively correlated with RSV disease severity, particularly the requirement for mechanical ventilation or longer hospitalisation [36,45,46]. A Type-2-skewed immunological environment, with a less robust Type-1 antiviral response, could explain the reason why those patients with more severe infection tend to be younger [28], especially with the recent discovery of an age-dependent, Type-2 response to RSV infection [30] (Figure 1). However, we are only just starting to understand how immune and epithelial cells communicate at the first interaction with a virus, as the balance between an efficient and damaging local inflammatory response can become dysregulated, especially in early life.

The primary aim of this review is to collate results on the mucosal response to RSV infection, including both immune and epithelial cell responses. This includes the production of cytokines in response to infection, but also the production of mucins, the major structural component of the mucus lining the epithelial surfaces in the airways and gastrointestinal tract, with their concentrations determining the viscosity of the mucus itself. We also investigate the evidence for immunoregulation in early life, characterised by signalling by immunoregulatory immune cells and Type-2 skewing. In addition, the discussion will focus on whether there are correlations between the Type-1 and Type-2 responses to RSV in the mucosa, with a focus on disease severity, and finally how it compares to responses seen in blood, and how it compares to other respiratory viruses. We highlight the importance of sampling the mucosa when studying RSV disease severity and exacerbation, the role of cellular immune responses, and immunosuppressive mechanisms.

## 2. Methods

To generate the studies for this literature review, an initial, systematic approach was employed, whereby an advanced literature search was performed through PubMed, with the final search being conducted on 27 October 2020, using the following search terms:

((infan*) OR (child*) OR (pediat*) OR (paediat*)) AND ((lung*) OR (airway*) OR (upper respiratory tract) OR (lower respiratory tract) OR (respiratory tract*)) AND (infant) AND ((RSV) OR (respiratory syncytial virus)).

These search terms were primarily used to capture all studies investigating immune responses to RSV in those under the age of two, whether this was measured in blood or the airway mucosa. Specific topics were followed up with additional literature, where the original search terms did not capture studies which were known to the authors, or which were found in paper references. For example, three studies [25,47,48] were added into Section 5 as these studies were published after the date of the literature search but supported the data originally collated from the relevant studies. There was a specific focus on human infant studies, but results from mouse models and in vitro studies were also included to support the results from human in vivo studies. This review defined “neonates” as those patients up to 28 days, and “infants” included children aged up to two years, though the definition of “infant” by age varied throughout the studies. Results were therefore included if the age range of participants in a study did not surpass two years of age. Once reviews and studies not relevant to this specific review were filtered out, the results were collated and organised by specific measurements, sample type, age-related differences in responses, and the response to RSV compared to other viruses. While the immune response of the respiratory mucosa was the primary remit of this review, studies of peripheral blood measurements were included where these complemented studies of the mucosa or filled knowledge gaps on mucosal immunity to RSV.

## 3. Results

### 3.1. Studies Included in the Review

After using the search criteria specified in the Methods section, the literature search returned 4572 studies which were then screened based on their titles and abstracts if they measured immune cell populations, cytokine and mucin concentrations, or RNA expression in the blood and/or upper respiratory mucosa during RSV infection. After excluding epidemiological or surveillance-based studies, those just using animal models, or investigations into vaccine candidates and testing, there were 79 studies that were relevant to the review. As also described in the Methodology section, some topics were followed up independently if the literature search did not capture all relevant studies, or if there were less evidence in human studies. Results from cell culture or mouse models of RSV infection were included in the sections pertaining to immunoregulatory cells during infection, as this has been less well-studied in infants. Furthermore, these important cell culture and mouse model studies can be evaluated to help direct future research conducted in vivo.

### 3.2. The Response of the Respiratory Mucosa to RSV Infection

There are several methods used to study the immune response of the airways following viral infection: aspirations (including nasopharyngeal aspirates (NPA) and tracheal or bronchial aspiration), nasosorption, and nasal wash/lavage (each aim to sample the mucous lining the airways and to focus on measuring mucosal-specific cells and immunological mediators); and nasal brush, micro-curette, and biopsies (although they primarily collect epithelial cells, they can also be used to study upper airway immune responses. The nasal mucosa is commonly preferred over tracheal or bronchial samples due to ease of sampling. Respiratory sampling in this manner can help determine tissue-specific responses at the site of infection. Nasal and respiratory sampling allow for the collection of the mucosal lining, including cytokines, mucins, and epithelial and immune cells, meaning mucosal-specific responses to RSV infection can be better understood (Figure 2). However, the degree to which URT sampling reflects the lower respiratory tract (LRT) is more difficult to determine, particularly as severe RSV disease is associated with LRTI and symptoms [49]. Evidence for cellular infiltration into the respiratory mucosa and its role in RSV disease has been investigated, including the discovery that neutrophilic infiltration prior to RSV infection predisposed adults to symptomatic infection [50]. In infants, the role of the mucosa and the immunological balance it maintains is essential, as it differentiates between pathogenic and commensal microbes and responds accordingly. As previously reviewed, the neonatal innate and adaptive immune responses are limited, but cells quickly mature as microbes infiltrate the airways [12]. Together, these studies indicate that the cellular composition of the airway influences the outcome of RSV exposure and the severity of symptoms.

#### 3.2.1. Evidence for Immunoregulation during RSV Infection

The environment of the infant respiratory mucosa is skewed towards tolerogenic responses, enabling colonisation of commensal microorganisms [12]. This immunosuppressive environment aids the microbial and structural development of the airway but may also alter responses to pathogens. There is growing interest in studying the cells driving immunoregulation during respiratory viral infection to understand whether they are protecting against damaging Type-1 inflammation or prematurely terminating the cellular response to infection and preventing viral clearance. We therefore need to carefully consider the roles of the innate and adaptive immune cells driving immunoregulation, such as alveolar macrophages and regulatory T cells/B cells, in the context of RSV infection. Although we are primarily interested in the mucosal response to RSV, the presence of these immunoregulatory cells in the airways is challenging and understudied in humans, particularly in early life. For this reason, studies using blood or murine models have also been evaluated to collate evidence on the role of these cells during infection before hypothesising how they behave in local tissues.

##### Alveolar Macrophages (AMφ)

AMφs provide an initial defence against pathogens that make their way into the respiratory tract [51]; these work alongside interstitial macrophages, which occupy the space between the alveolus and vasculature, to help coordinate an efficient immune response with help from epithelial and other immune cells [52]. Two major subtypes of macrophages (“M1” and “M2”) have been described, which are phenotypically and functionally distinct. M1, or “classically activated” macrophages are primarily pro-inflammatory and are responsible for the phagocytosis of pathogens and subsequent release of inflammatory mediators. By contrast, M2, or “alternatively activated” macrophages have regulatory functions, working to subside the inflammatory response [53] and encourage tissue repair. At steady sate, AMφs were shown to express M1 and M2 markers simultaneously, and in the context of HIV infection, expressed significantly less of the M2 marker CD163 [54]. While the M1/M2 paradigm is a convenient nomenclature, these cell types likely represent the polar ends of AMφ phenotypes, which may be better described as a spectrum between these two extremes in vivo [55]. Studies in murine models have identified that the Type-2 cytokines IL-4 and IL-13 are potent inducers of M2 macrophages [56]. IL-10 has also been shown to enhance M2 polarisation by increasing the expression of IL-4 receptor (IL-4Rα) expression in vitro [57]. Therefore, communication between the Type-2 response and M2 macrophages is necessary to limit inflammatory damage to the tissues lining the respiratory tract. The polarisation of airway macrophages in early life and how this might affect the neonatal/infantile response to inhaled pathogens are understudied. As previously mentioned, AMφs are more likely to polarise into the M2 phenotype under the influence of IL-13, which can be derived from ILC2s, which subsequently limits their phagocytic activity and inflammatory responses [26]. In a Csf1r-EGFP reporter mouse model, postnatal lung development was strongly associated with the polarisation of macrophages into the anti-inflammatory M2 phenotype, as measured from consistently high expression of *Arg1, Ccl17,* and *Mrc1*. This upregulation peaked at day 14 following birth before returning to lower levels in the adult lung [58]. Favouring this M2 phenotype could be a protective mechanism against extensive inflammation that could occur following birth and the infiltration of air pollutants, possibly to support the extensive remodelling of the lung during alveologenesis. In the context of RSV infection, isolated AMφs secrete a variety of cytokines, including IFN-α, IFN-β, CXCL10, TNF-α, and IL-6 [59], showing that they present a primarily inflammatory phenotype in vitro. AMφ polarisation has not been measured in the respiratory samples of infants infected with RSV, and most results presented have been measured from murine models or in vitro studies reflecting the difficulty in studying these cells in vivo in human infants. In a study measuring AMφs in the bronchoalveolar lavage (BAL), it was shown that AMφ immunophenotypes were very similar between infants and adults through equivalence in the expression of M1 and M2 activation markers. However, chemokine production by infant AMφs was altered compared to adult AMφs in culture supernatants following *Mycobacterium tuberculosis* infection, suggesting a difference in AMφ-mediated immune signalling in early life [60]. Specifically, decreased expression of chemokines that signal through CXCR3 was measured in infant AMφs. These chemokines, including CXCL9, -10, and -11, all function to recruit Th1 cells, suggesting there could be a reduced Type-1 response by infants during infection. Determining whether these AMφs favour a more Type-1 or Type-2 phenotype in a healthy state or following RSV infection, and how this is impacted by environmental factors, would be crucial to understanding how AMφs impact early life respiratory immunity.

##### Regulatory T Cells (Tregs)

Tregs are a major branch of the immunosuppressive response, characterised by the production of IL-10 and the regulation of inflammation. The primary difference between neonatal and adult T cell populations is the higher expression of FoxP3 following TCR stimulation in neonates. FoxP3 is a transcription factor required for the development of Tregs [61,62], functioning to the suppress NF-κB and inflammatory cytokine production [63], thereby mediating the immunoregulatory phenotype. Activated (aTregs) and resting (rTregs) Tregs are two commonly defined subtypes in humans, defined by the differential expression of CD45a, FoxP3, or CD25 [64,65]. Tregs contribute to immune tolerance and prevent foetal rejection [66] throughout pregnancy, with these Tregs being transferred onto the neonate. This continuous transfer of Tregs to the foetus may explain why neonates born pre-term have significantly lower frequencies of CD4^+^CD25^hi^FoxP3^+^ Tregs than do term neonates [67]. As Tregs function to maintain homeostasis and downregulate T cell proliferation [68], reduced concentrations could result in exaggerated and damaging inflammatory immune responses to infection in those born prematurely. The importance of Tregs in RSV disease has also been shown in murine models, where abrogating Tregs prior to infection led to greater disease severity, enhanced CD4/CD8 T cell responses, and lung eosinophilia compared to WT mice [69]. The persistence of Th2-type responses with eosinophilia, and significant weight loss, was measured at the beginning of days 6 and 7 post-infection, respectively. This therefore highlighted the role of Tregs in preventing this longer-lived pathological Type-2 response to RSV infection.

In the peripheral blood of RSV-infected infants, only aTregs (CD45RA^lo^CD25^hi^) were significantly reduced compared to healthy controls [64]. This was matched with increased IFN-γ, TNF-α, and IL-4 production, suggesting that there was reduced aTreg-mediated downregulation of T cell signalling. This is consistent with other observations of reduced Treg populations in the peripheral blood of RSV-infected infants compared to healthy controls. This was only the case for rTregs, aTregs, and FoxP3^+^/CD4^+^ Tregs (not naive or memory populations), which could suggest a reduction in the blood counts of functionally active Tregs at the time of infection [70]. In another study sampling blood and PBMCs throughout the course of acute infection, days 0–8 post-symptom onset, the rTreg concentrations significantly declined [64]. The decrease of rTregs could be due to their conversion to aTregs [65] to function in local tissues to promote tissue repair as infection resolves, as aTregs were not shown to increase systemically [64]. Repeated respiratory sampling throughout acute infection would help to determine whether these aTregs play a role in resolving inflammation and fixing the epithelial barrier following infection. In local respiratory tissue, specifically in the upper airways, there was an age-dependent, significant increase in the counts of Th17 cells, but a significant decrease in Treg frequencies in participants ranging 2–94 years [71]. The significant reductions in blood Treg counts during RSV infection could also suggest migration to the airways, meaning there could be functioning Tregs in the nasal mucosa in very early life to combat RSV-induced damage or immunopathology, as previously shown in mice [72]. Although this work is important to understanding the role of Tregs in the systemic compartment, or in murine models of RSV infection, determining the functionality of Tregs in the airway mucosa in infants would help to determine these potential protective mechanisms in vivo.

##### Regulatory B Cells (Bregs)

Regulatory B cells, or Bregs, also possess immunosuppressive activity. They function in a similar manner to Tregs, whereby they release regulatory cytokines, predominantly IL-10 and IL-35, to dampen inflammation and prevent an overexuberant immune response. For example, in reporter mice, IL-10-producing B cells were able to suppress the production of pro-inflammatory cytokines and protect against autoimmunity, highlighting their regulatory function [73]. In humans, the Breg cell subtypes are split between B10 cells, plasmablasts, immature cells, and Br1 cells. Key features of all these subtypes include their abilities to produce IL-10 and suppress DCs or CD4^+^ T cells, as previously reviewed [74]. More recently, Bregs identified in the cord blood and both the blood and NPAs of neonates (termed “neonatal Bregs”, or “nBregs”) have been identified and immunophenotyped using mass cytometry techniques. These cells express high levels of CD5, which differentiates them from other cord-blood-derived B cells [27]. Unlike Tregs, Breg concentrations were not significantly different in the cord blood of those born pre-term or at full-term [27]. However, all B cell populations waned quickly with age, suggesting that the risk of nBreg cell-enhancement of RSV disease severity declines through infancy. These Bregs are also not present in the adult lung, highlighting the importance of studying early-life immunity. A neonatal BALB/c mouse model of RSV-infection showed that Bregs colonise the lungs in the first week of life and can affect the overall respiratory immune environment [75]. These neonatal Bregs dampened the release of IFN-γ in response to RSV infection in an IL-10-dependent manner, resulting in a failure to contain RSV replication. This has been corroborated in human studies, where it was shown that these nBregs can be infected with human RSV (hRSV), and subsequently dampen the Type-1 immune response, promoting a biased Type-2 response and worsening disease outcomes [27,76]. An alternative method that has been proposed for nBregs to worsen RSV-associated pathogenesis is through the suppression of AMφ antiviral function [75]. The IL-10 produced by nBregs can suppress the secretion of IFN-α and -β ex vivo, influencing the AMφs towards a Type-17-like phenotype instead. These results from in vitro, ex vivo, and murine models present a possible role for nBregs in the greater susceptibility of neonates to severe RSV infection.

#### 3.2.2. Resident and Infiltrating Immune Cells to the Respiratory Mucosa during Infection

Various immune cell types reside in the airway, including AMφs, DCs, neutrophils, ILCs, eosinophils, and tissue-resident memory (Trm) T cells (Figure 2). Trm cells in the lungs have been detected in both infant and adult mice and have shown great capacity for protection against viral re-infection [77,78]. In human paediatric participants, an age-related maturity of Trm cells suggests increased T-cell mediated protection in the lungs in later childhood, with depleted populations in very early life contributing to immature immunity [79]. Immune cells migrate to the lung from the alveolar blood supply, and infiltration of these cells into the lung interstitium, parenchyma, and lumen of the airways contributes to local immune responses. Changes in the immune cell populations of the airway during RSV infection may reflect migration to the mucosa and be mechanistically associated with disease manifestation and severity.

##### Immune Cells Measured in Respiratory Samples of RSV-Infected Infants

Frequencies of both myeloid and plasmacytoid DC (mDC and pDC) have been shown to significantly increase during RSV infection, compared to healthy controls, as measured in nasal washes from infants less than 6 months of age [80]. The increased nasal frequency, but decreased blood frequency, of DCs suggests migration from the blood to enact a local immune response to RSV. Finally, the frequency of monocytes, CD4^+^ T cells, and CD8^+^ T cells in nasal washes were not significantly different between RSV-infected and healthy control groups [80]. Flow cytometric analysis of NPAs revealed that both the frequency and absolute counts of ILC2s were significantly increased in a severe (PICU-admitted) RSV-infected disease cohort compared to a moderate severity (non-PICU-admitted) group [30]. ILC2s release IL-13 and trigger migration of activated DCs to the lymph nodes and stimulate the differentiation of Type-2 cells from naive CD4^+^ T cells [81]. Therefore, increased frequencies of ILC2s in severe RSV support earlier studies suggesting a biased Type-2 immune response during severe RSV disease [26]. In a study investigating specific CD4^+^ and CD8^+^ T cell populations in the nasal aspirates of RSV-infected infants, Type-2 skewing was associated with more severe disease. Specifically, the frequencies of CD3^+^/CD8^+^ T cells or cytotoxic T cells (Tc1 or Tc2) were significantly higher than CD3^+^/CD4^+^ T cells, which matched with a significant Tc2/Tc1 ratio and frequencies of IL-4^+^/IFN-γ^+^ T cells [82]. All of these were significantly associated with severe RSV infection, suggesting that local Tc2 skewing drives RSV pathogenesis. As this Tc2-specific response has been shown to arise 5–9 days following hospital admission, and Th2-associated cytokines have been detected earlier in more severe infection [30], it could be hypothesised that a greater initial Th2 response induces a more robust Tc2 response and leads to a worse clinical outcome. Contrastingly, intracellular cytokine staining performed on nasal brush samples showed there were no significant differences in the frequencies of cells positive for IL-10, IL-4, and IFN-γ between RSV-infected severity groups [83]. This could be due to a difference in the sampling technique, with nasal brushings being the preferred technique for the collection of epithelial cells [84,85]. This is reflected in another study that undertook bulk-transcriptomics of nasal brushings and blood samples collected from RSV-infected infants amongst those with asymptomatic rhinovirus or healthy controls [86]. Blood samples from infected participants showed an increased signal for all immune cell markers, whereas, unsurprisingly, nasal brushings were enriched with marker genes for epithelial cells and mucins. In the nasal brush samples, however, a heightened difference in leukocyte markers in those with RSV infection, compared to controls, could be suggestive of immune cell migration to the nasal mucosa.

Comparing immune cell populations in nasal brushings from infants with either RSV-induced upper respiratory tract infection (URTI) or bronchiolitis showed comparable concentrations of macrophages and T cells [83]. As infection resolved, significant decreases in macrophages, T cells, and eosinophils were observed. However, without comparing these cell counts to uninfected controls, it was not clear how different the counts of immune cells were in the acute or convalescent phase to an age-matched healthy baseline [83]. In RSV-positive infants with bronchiolitis, there was a reduced expression of STAT3 compared to those with bronchiolitis due to other respiratory viruses [87]. As STAT3 is an important mediator for CD4^+^ T cells, CD8^+^ T cells, and B cell activation and proliferation [88], this was investigated in the context of RSV-induced bronchiolitis [87]. It was determined that RSV interfered with the IL-21-mediated phosphorylation of STAT3 in memory CD8^+^ T cells and resulted in reduced cytotoxicity against RSV-infected epithelial cells. Although the mechanistic work was done in cell lines and memory CD8^+^ T cells collected from healthy adults, it explains a potential mechanism for reduced T cell responses in paediatric RSV infection. Finally, overall immune cell counts in either the blood or upper-airway aspirates of children under the age of four were evaluated, whereby neutrophils, but not monocytes or lymphocytes, were significantly different in the blood of those with respiratory viral infection, compared to uninfected controls [89]. In airway samples, however, the proportion of neutrophils and monocytes present were not different between infected and uninfected participants, but CD3^+^ T cells were significantly different. Investigating this further showed a significant increase in both CD4^+^ and CD8^+^ T cells in the infected group, with the CD8^+^:CD4^+^ ratio being skewed even further in the infected group with those that developed acute lung injury [89]. This could therefore be a marker of persistent inflammation, associated with more severe disease, at the site of infection. However, the grouped viral infections mean it is unknown whether this same immunological and clinical phenotype is mirrored in RSV infection alone. Although both measurements were performed, immune cell populations in the blood and airways were presented in such a manner that direct comparisons could not be done, but heterogeneity in the immune-cell compositions between cohorts was evident in both the blood and airways, highlighting the importance of understanding the presence and function of immune cells in both systemic and mucosal compartments.

Collated results from studies investigating the mucosal cellular response to RSV suggests that it is primarily DCs and ILCs which increase in frequency during infection. To better characterise overall cell populations, flow cytometry combined with -omics techniques could help to identify both cell type frequencies and their specific phenotypes. Single-cell RNA-sequencing (scRNA-seq) techniques can also allow for the phenotyping of individual cell types from local tissues and can provide insight into their function at the point of infection, as well as how they recover post-infection. What we also need to consider, is how RSV is modulating the function of these cells and how it can impact the subsequent immune response. Ex vivo culturing of monocytes, using GM-CSF and IL-4, produced immature DCs that were phenotypically consistent with usual immature DC characteristics [90]. Infection of these DCs with RSV lacking the NS1 protein (ΔNS1) had enhanced IFN-α and IFN-β production compared to whole virus infection and other RSV knockouts. This study found that the NS1 protein is an antagonist of type-1 interferons and many other cytokines ex vivo, and impaired DC maturation. This could therefore be a mechanism for reduced DC function and antigen presentation to CD4^+^ T lymphocytes during RSV infection and could potentially be hindering the adaptive immune response.

#### 3.2.3. Immune Cell Stimulation as a Measure of Functionality

Leukocytes collected from the respiratory tract are commonly not suited to functional analyses, where cells are required to persist in tissue culture environments. To overcome this limitation, some studies have utilised cells from peripheral blood and extrapolated these results to the potential activity of the equivalent cell population in the airway. This approach is a limitation to such studies as comparability with airway populations cannot be assumed, but it offers a pragmatic tool for studying cellular function.

To identify how immune cells are functioning in response to infection, mechanistic studies have performed LPS- or phytohemagglutinin-stimulation of immune cells of interest, collected from either infected or healthy participants and measuring cytokine responses. In infants with confirmed RSV-infection, this has almost exclusively been performed in blood, where PBMC stimulation can provide insight into the blood responses to viral infection [45,91,92], but have shown few differences in cytokine production following stimulation. Another hypothesis is that immune cell perturbations (or differences in the skewing of the immune response) precede infection, driving differences in early immune responses, which has been recently shown to be the case in adult RSV infection [50]. Prospective birth cohort studies have been conducted to study this possibility in early life, where predictive regression analyses commonly enable such studies to understand predisposition to infection and disease severity. Immune cell and cytokine concentrations in the venous blood of one-month-old infants were obtained for prospective measurement. The primary differences in systemic immune cell populations were only in the number of eosinophils and monocytes, which were increased in the blood of infants who later developed RSV LRTI. Concentrations of DCs, neutrophils, and basophils were not significantly different, but the functional maturity and tissue-homing potential of these, and many other cell types, are understudied in relation to respiratory viral disease severity. Finally, in another prospective study measuring cytokine concentrations in cord blood, IL-12 was significantly lower in new-borns who later developed RSV-induced bronchiolitis [93]. As a potent inducer of Type-1 responses, a decreased IL-12 concentration could also be indicative of a weaker Type-1 response as a predictor of infection.

Cord blood mononuclear cell (CBMC) isolation is another ex vivo model to measure the efficiency of the immune responses at birth in infants who either remain healthy or subsequently develop disease. In LPS-stimulated CBMCs, logistic regression analysis demonstrated that a combination of increased IL-6 and IL-8 predicted severe RSV disease before 6 months of age, compared to infants who were treated as outpatients or who remained healthy [94], which corroborates a cytokine analysis of plasma [32]. IL-4 and IFN-γ release by unstimulated CBMCs significantly increased in infants subsequently hospitalised with RSV [94]. Using CBMCs and pre-infection peripheral blood samples provides important insight into the potential role for immune cells in susceptibility to RSV. Identifying immunological differences is fundamental to understanding how this impacts susceptibility to infection, as well as the clinical outcomes of the disease itself, but in infants and young children, this is extremely difficult to conduct.

## 4. Cytokine Production in Response to Early-Life RSV Infection

Although we have seen differences in immune cell populations in RSV infection compared to healthy controls, as well as a link with disease severity and age of infection, cytokine production and detection can also elucidate the immunological pathways activated and how this affects clinical outcomes. Several factors must be taken into consideration for cytokine analysis. High levels of inflammatory mediators reported in many studies may associate with correspondingly robust anti-inflammatory responses. Alternatively, immune responses may be skewed toward inflammatory (Type-1 or -17) or anti-inflammatory (Type-2) responses. As already described, a skewed Type-1 response could result in long-term damage to the airways, but favouring a Type-2 response could prematurely resolve the antiviral response [95] and delay viral clearance. The relative roles of Type-1, Type-2, and Type-17 responses in the mucosal and systemic immune responses to RSV infection have been investigated through measurement of cytokine concentrations in RSV-infected infants. For upper respiratory tract (URT) sampling, both nasosorption [96] and NPAs provide comparable measurements of viral quantities, which is important in a clinical setting, as they are easier-access diagnostic techniques [28,38]. The quantification of cytokines and immune mediators is also typically logistically easier than cellular analysis and allows a broader array of immune factors to be studied than permitted by most techniques. Analysis of cytokines can also allow interpretation of immune response skewing (e.g., Type1/2), but cannot distinguish the cellular sources of these immune mediators.

### 4.1. Upper Respiratory Sampling Using NPAs and Nasosorption

In upper airway samples, significant increases in IFN-γ, TNF-α, and IL-4 in RSV-infected infants, compared to healthy controls, suggests an increased activation of both Type-1 and Type-2 pathways [64,97]. Increased concentrations of IL-4, IL-6, IL-13, IL-33, and IL-10 in RSV(+) NPA samples could discriminate between mild and more severe disease [30,36,98,99], indicating that there could be partial Type-2 skewing in the respiratory mucosa during enhanced RSV pathogenesis. Another indication of Type-2 skewing can be measured by quantifying the ratio of Type-1/Type-2 cytokine concentrations as a direct comparison. Biased Type-2 responses, characterised by increased IL-4/IFN-γ or IL-10/IL-12 ratios, are not only significantly increased in RSV compared to healthy controls, but are also associated with disease severity [36,98]. This agrees with what has been measured in the plasma/serum of infants with PCR-confirmed RSV infection, where significant increases in IL-6, IL-10, IL-12, IL-13, and IL-35 levels have been reported compared to controls, indicative of a robust Type-2 response [32,39,93,100]. In nasosorption samples collected from RSV-infected infants, a decreased IFN-γ and CCL5/RANTES expression coupled with exaggerated IL-17 expression and mucogenesis were markers of more severe bronchiolitis [28,101].

When utilising human samples to study acute infection, one factor to consider is the timing of recruitment and sample collection. The majority of studies included in this review obtained respiratory samples from their participants a few days post-symptom-onset, or within 24–48 h of hospital admission. However, one study measured cytokine levels in nasal aspirates at discharge, which represents a different immunological environment from admission, as infection is resolving [102]. In this study, both Type-1 and Type-2 cytokines were significantly higher in the RSV infection group, corroborating studies investigating earlier stages of infection, but there were also significant levels of cytokines important for macrophage and eosinophil induction, as well as markers of tissue repair and recovery (VEGF, PDGF, and G-CSF [103]). Furthermore, the cytokine concentrations at discharge compared to one-year post-infection were measured, and cytokines were grouped based on their expression pattern at the two time-points. IL-10, IL-7, and IFN-γ all maintained a stable expression over time and could be playing a role in longer-term immunity, but IL-1RA, MIP-1α, IL-4, TNF-α, and more were all significantly decreased one-year post-infection compared to levels at discharge. This therefore suggests that these cytokines are particularly important during acute infection and resolution of the virus, but it is not certain how quickly they waned post-infection. Another factor to consider is the differences in results between sample types, as some techniques may collect different immunological mediators. For example, in a study that measured cytokine concentrations in both NPA and nasosorption, only the results from nasosorption were statistically different between cohorts [38]. Overall, these studies do not support a strong biasing towards either Type-1 or Type-2 cytokine responses in the mucosa in RSV-infected compared to healthy controls, but instead show an overall increase in both pro-inflammatory and regulatory cytokines during infection. However, when studying high severity groups, there is a potential for Type-2 skewing associated with greater immune dysregulation.

Age-related differences must also be considered, particularly in longitudinal studies of infants who developed RSV infections at different developmental stages. In healthy post-natal development, Type-2 skewing has been measured in a neonatal mouse model, where it was demonstrated that the release of IL-33 by epithelial cells immediately following birth is sufficient to trigger a Type-2 signalling cascade [26]. This subsequently promoted ILC2 recruitment and macrophage polarisation toward an immunoregulatory phenotype [26]. In another neonatal mouse model, RSV infection was shown to rapidly induce high levels of IL-33 and IL-13, and increased concentrations of ILC2s were also measured post-infection [104]. Blocking IL-33 in this model was sufficient to reduce RSV immunopathology, highlighting this pathway in driving RSV disease severity in early life, but it is not known whether this occurs in human neonates, or for how long this phenotype persists. In human infants, one study found that the Type-2 skewing associated with disease severity was heavily dependent on age [30]. Type-2 cytokines and the concentrations of ILC2s were significantly greater in those younger than 3 months of age, but IFN-γ was increased in the older cohort. This has also been measured in plasma, where age-dependent increases in the concentration of IFN-γ were measured with confirmed RSV infection [100]. In a whole-blood transcriptomics assay in infants hospitalised with RSV lower respiratory tract infection (LRTI), interferon signalling was significantly upregulated in RSV, but immune dysregulation was also observed. B cell signalling and development, Th cell differentiation, NF-κB signalling, and IL-4/IL-10 signalling were all significantly downregulated in RSV LRTI, and this immune dysregulation was more pronounced in infants less than 6 months old [105]. As there is evidence for increased IL-4 and IL-10 responses in the airways of RSV-infected patients, the reduced blood transcriptomic markers could be due to a local response to RSV, but with limited evidence for immune cell signalling in the airways during infection, this is still unknown. Furthermore, the rapidly changing nature of neonatal and early life immunity means that even small differences between the demographics of patient populations in disease and control groups may account for major differences in the apparent immunology of disease.

### 4.2. Differences in Cytokine Profiles in the Upper and Lower Airways

There have been fewer studies measuring the immune responses of the lower airways, which is typically performed using bronchoalveolar lavage (BAL), bronchial aspiration, or bronchial biopsy. When analysing the results from the studies measuring the cytokine responses in BAL specifically, levels of TNF and IL-6 protein were significantly higher in RSV bronchiolitis compared to those in healthy controls, and this difference persisted ~5 days following initial sampling [35]. In a different study, elevated levels of Type-1, and -2 cytokines, as well as other chemokines and growth factors, were all measured in RSV bronchiolitis patients relative to healthy controls, apart from IL-12p70, which was non-significantly increased in controls, and IFN-γ and IL-17 were not detectable in either cohort [106]. IL-12p70 was the only cytokine decreased in RSV bronchiolitis. Although it was not statistically significant, when combined with undetectable IFN-γ, it was hypothesised that RSV was modulating cell-mediated immune responses through IL-12p70 in the lower airways, where IL-12p70 is another important cytokine for Th1 differentiation from Th0 cells [107]. NPAs were also collected from the same participants, but a much smaller response to RSV was measured, with lower expression levels of cytokine mRNAs in RSV-infected subjects compared to healthy controls. Therefore, a different cytokine profile was detected in the NPAs, but both NPA and BAL exhibited significantly higher CCL2, a broad myeloid cell attractant, in RSV-infected subjects compared to healthy controls [106]. Although both sample types were collected, cytokine expression was measured using different methods, so statistical analysis could not be performed to compare airway mediator levels between the BAL and NPA. This comparison would begin to address a knowledge gap in the differences in the response to RSV in the upper and lower airways.

### 4.3. The Anti-Inflammatory Function of IL-10

We have already described potential roles of immunoregulatory cells in early life and RSV infection, but what we also need to consider is whether the cytokine responses reflect an increased immunoregulatory response, and whether it is indicative of clinical outcomes. IL-10 is the most studied immunosuppressive cytokine in RSV-infected infants. The role of IL-10 in RSV is still under speculation, as there is evidence for it being protective against severe clinical outcomes, but it has also been linked to severe disease due to the premature suppression of the antiviral response. When comparing RSV cases to healthy controls, IL-10 was found to be significantly increased in infected infants [28,64,83,97,98,99,102,106,108,109] and positively associated with viral load but not disease severity [38]. However, some studies have identified differences in IL-10 concentrations between symptom severities; for example, IL-10 was significantly increased in the NPAs of the acute bronchiolitis (AB) group compared those who developed a more moderate URTI [34]. Contrastingly, IL-10 was not significantly increased in the acute bronchiolitis (AB) cohort in the first few days of infection before declining to below the URTI cohort levels [36]. In this same study, decreased IFN-γ and increased IL-4 in infants with AB were found to be suggestive of a Type-2 biased response, which was also measured in the form of significantly increased IL-10/IL-12 ratios in the AB group compared to the URTI cohort [36]. Given the immunosuppressive role of IL-10 and the correlation with viral load, infants who develop more severe disease could be faced with persistent viral replication and pathogenesis that rises from an uncontrolled virus. The enhanced tissue damage that is enabled by raised levels of IL-10, potentially through favoured Type-2 responses, in upper respiratory samples is thought to be one mechanism that increases the likelihood of developing post-bronchiolitis wheeze [95].

### 4.4. Differential Immune Responses to RSV Compared to Other Respiratory Viruses

There are several other viruses which are responsible for upper and lower respiratory tract infection in infants; rhinovirus (RV), parainfluenza virus (PIV), human metapneumovirus (HMPV), coronaviruses (HCoV), and human bocavirus (HBoV) range not only in their prominence during seasonal outbreaks, but also in the severity of symptoms and the amount of time that symptoms persist post-infection [110,111]. Identifying differences in the immune responses between RSV and these other respiratory viruses may yield important information on pathogen-specific mechanisms of disease. For this review, studies were included if they measured the cytokine responses to RSV and another respiratory virus as separate cohorts. There were three studies that measured the respiratory cytokine responses to RSV compared to either HMPV, RV, or HBoV.

Nasal secretions of RSV- or HMPV-infected infants did not show any statistically significant differences between cytokine concentrations, but the results suggested increased IL-10 and IFN-γ levels in RSV-infected full-term and premature infants compared to those infected with HMPV [98]. In infants infected with either RSV or RV, there were differing cytokine responses evident between the two virus groups, as well as versus healthy controls. Cytokine and miRNA measurements from nasal brush samples of patients demonstrated increased concentrations of IL-10 and IL-13 in infants with RV (*p* < 0.05), and both viruses had differential effects on NF-κB signalling [112]. Infants with RSV infection had a miRNA profile that the investigators predicted would downregulate NF-κB expression, but RV infection upregulated expression, based on the expression of 84 genes associated with the signalling pathway [112]. However, patient ages ranged from 2–12 months, and so it is uncertain whether there is a contributing age-dependent change in cytokine production or miRNA profile that is not necessarily seen in neonates or younger infants, or vice versa. A prospective birth cohort study was designed to capture the immunological profiles of first infection with either RSV or RV [113]. Although there was some overlap in the mediators enriched in infections with either virus, mediators associated with viral clearance, such as CXCL10, CXCL9, CCL11, and CCL2, were preferentially decreased in RSV infection. The primary immune responses to RV, however, were more associated with growth factors. In those with recurrent wheeze following RSV infection, Type-2 and Type-17 responses were significantly enriched, showing an overall difference in the immune responses between the two viruses.

Finally, nasopharyngeal cytokine concentrations have been compared between infants less than 2 years of age infected with either RSV or HBoV, and healthy controls. All cytokines, apart from IL-5, were increased in concentration in virally infected infants compared to controls, while RSV-infected individuals had increased IL-10 and TNF-α concentrations relative to HBoV, but a decrease in IFN-γ [97]. Across all the airway-specific cytokine responses to respiratory viruses compared to RSV, RSV exhibited increased IL-10 concentrations, suggesting greater regulation of the immune response that could subsequently lead to reduced viral clearance. However, one study measured no significant differences in all cytokines measured in the nasal secretions of infants with either RSV or a different viral infection, irrespective of the clinical manifestation of the infection [34]. The age of infection does need to be taken into account, as the age at RSV infection is commonly younger than with other respiratory viruses, and the same can be said for disease severity [94]. There are therefore two factors that need to be explored further: the immunopathology of RSV compared to other viruses, and the changes to the immune environment of the airways throughout healthy infancy that could provide advantageous conditions for RSV infection in earlier life. One limitation to comparing the immune responses to different respiratory viruses is the differences in their clinical presentations, as RSV infection tends to present more severely than other viruses, in particular compared to RV infection [114].

## 5. Epithelial Cells Mediating the Immune Response to RSV

As the primary site for RSV infection, it is important to also study the response of infected epithelial cells and how their downstream signalling mediates immune cell recruitment to the airways. The primary ways to measure the epithelial cell responses to RSV are to measure the transcriptional profiles of ex vivo epithelial cells, 2D or 3D epithelial cell cultures, or mouse models. Three-dimensional cell cultures are increasingly popular, comprising the development of air-liquid interface (ALI) cultures to produce a differentiated epithelium to best mimic the conditions in vivo. For these cultures, site-specific epithelial cells, for example nasal-, bronchial- or tracheal-derived epithelial cells, can be cultured to recapitulate the cellular environment of the upper or lower airways [115,116,117,118]. Basal cells, which are the progenitor cells of the airways, can also be used to grow cultures of a differentiated epithelium from a more naive state [119]. In human airway epithelial cells (HAECs), CX3CR1 expression by ciliated cells was essential for RSV G protein binding and infection [120]. In well-differentiated paediatric bronchial epithelial cell (WD-PBEC) cultures obtained using bronchial brushings from RSV-infected infants, IL-29 was found to be important for driving an RSV-induced antiviral state in vitro [121]. RSV infection was also found to affect the differentiation of an ALI culture model of epithelial injury in vitro, measured in a study that tracked basal cell differentiation and epithelial cell phenotypes up to 21-days post-infection [122]. In the basal cell cultures that were infected with RSV, the differentiated epithelium showed a significant loss of ciliated epithelial cells, but an increase in secretory cell formation matched with mucin immunostaining (MUC5AC/MUC5B specifically), compared to the uninfected cells [122]. This therefore showed that RSV infection was able to influence the cellular composition of the airways by hijacking the proliferation of basal cells, to promote goblet cell formation, and the production of mucus via mucin expression.

### Evidence for the Immunogenicity of Mucins

Mucins make up part of the structural component of the respiratory mucosa; they are a group of glycoproteins incorporating complex carbohydrate chains that add to the heterogeneity of their structure and function [123]. They are split into cell-surface and secreted mucins, with the latter being further categorised into gel-forming and non-gel-forming [124]. As previously reviewed, changes in mucin composition during viral infection provide an explanation for the physical changes to the mucus, where a reduction in mucociliary clearance and subsequent airway obstruction is associated with RSV infection [125]. Alongside their role in mucociliary clearance, mucins have pro- and anti-inflammatory roles in response to bacterial or viral infection [126]. Several mucins are abundant throughout the airways, potentially contributing to the innate immune responses to RSV [127]. Mucin overproduction has been measured in RSV-infected BALB/c mice, but increased production could only be detected in the airway epithelium infected with RSV strains 19 and 2–20, not A2 and 3–12 [128]. Differences in mucogenic responses to subtly different RSV strains and the subsequent effect this has on mucous composition and viral clearance could explain some of the variance in RSV disease severity. Corroborating the immunogenic effects of mucins in the airways, *Muc5b-/-* mice not only had significantly reduced mucociliary clearance in the upper and lower airways, but also had significantly increased mortality rates compared to wild type (WT) mice [129]. The increased mortality rate was directly due to the infection caused by pathogenic bacteria, not just because of reduced airway clearance and potential blockage. *Muc5b*-depleted mice also had altered immune cell profiles throughout the airways: lymphocytes were absent from the bronchoalveolar lavage fluid (BALF), macrophage phagocytic activity was severely reduced, and lung inflammation was resolved at a significantly reduced rate, providing a possible explanation for the elevated fatality [129]. To identify potential mucin expression changes during RSV infection, cell culture studies measured the differential expression of mucin RNA by RSV- and human-metapneumovirus (HMPV)-infected A549 epithelial cells [127]. Both secreted- and cell-surface mucin gene expression was measured at 12, 24, 48, and 72 h post-infection to detect the rate at which mucin induction takes place. Most secreted mucins showed a significant upregulation as a result of HMPV infection, with concentrations surpassing those caused by RSV. RSV infection caused a significant induction in the expression of MUC8, MUC4, MUC15, MUC20, MUC21, and MUC22, as well as a non-significant increase in MUC5AC expression compared to uninfected A549 cells [127]. In a similarly designed study, A549 cells infected with RSV A2 produced high levels of MUC1 in a dose-dependent manner; subsequent findings included the suppression of TNF-α by MUC1, suggesting an anti-inflammatory role for this mucin during RSV-driven inflammation [130]. In human bronchial epithelial cells (HBECs), RSV caused downregulation of expression of miRNAs hsa-mir-34b/c-5p, which are important regulators of MUC5AC production, suggesting another potential mechanism by which RSV induces mucin-associated mucogenesis [131].

Other studies have analysed mucogenic responses in respiratory samples of RSV-infected infants. We previously demonstrated that moderate RSV disease was associated with higher VL and IL-10 levels, whereas IL17A and MUC5AC were two mediators that had increased expression in infants with more severe LRTI [28]. MUC5AC levels were corroborated in another study when comparing mild, moderate, and severe disease groups. Levels were significantly increased when comparing cohorts with increasing disease severities [132], but it is not known whether this was matched with decreased mucociliary clearance, due to difficulty in determining this. However, it does show that severe RSV disease is associated with physical changes to the mucus, as increases in a gel-forming mucin MUC5AC likely results in increased mucus viscosity and airway obstruction. A link between Type-2 responses and these mucins has been proposed, suggesting that IL-13 indirectly induces MUC5AC production, leading to decreased mucociliary clearance [133]. The mechanisms and pathophysiology behind mucins and mucus production needs to be better understood. It could be that increased mucus production is advantageous to the virus, in terms of increased capacity for infection or migration to the lower airways, or aids in droplet transmission.

## 6. Genetic and Environmental Imprinting on Mucosal Immunity

The commensal microbiome develops from birth, and its evolution in the airways and GI tract is influenced by a multitude of environmental factors, including the route of delivery, use of antibiotics, and household type and location, as previously reviewed [134]. In one example, the beta-diversity (composition of bacterial species) in the infant gut was significantly associated with the presence of siblings, type of house, and use of antibiotics during pregnancy [47]. There is also evidence for breastfeeding contributing to the microbiota measured in NPAs, where increased presence of *Dolosigranulum* and *Corynebacterium* in breastfed infants could be a protective mechanism against respiratory infections [135]. Previous 16S sequencing of respiratory or gut microbiomes has shown the role of the colonisation of specific bacteria, including OTUs associated with *S24_7*, *Clostridiales*, *Odoribacteraceae, Lactobacillaceae,* and *Actinomyces*, in impacting respiratory immunity. There is also an age-related difference in the dominant bacterial species in the upper airways [136]. At 2 months of age, the most abundant bacterial phyla included those belonging to *Staphylococcus* and *Corynebacterium*, which were then replaced by *Allolococcus*, *Moraxella*, *Streptococcus,* and *Haemophilus* by roughly 6 months of age. What is not known, in regard to the healthy airways, is whether these bacterial species correlate with levels of immune cells or mediators that would influence the responses at the start of an infection. In the nasotracheal aspirations of RSV-infected children, it has been shown that the abundance of specific bacterial species positively correlated with mucosal cytokine concentrations. For example, *Haemophilus, Moraxella,* and overall LPS levels in the airways positively correlated with CXCL8, IL-6 and IL-10, and IL-17A, respectively [137]. Therefore, bacterial colonisation does affect the immune environment of the airway. Bacterial profiles from both the airway and gut microbiota are distinct in RSV-infected infants compared to healthy controls [85,138,139]. These microbiomes are also distinct between disease severity groups, where more severe disease is associated with a dominance of *H. influenzae*, *Moraxella*, *Streptococcus, Alphaproteobacteria*, *Gammaproteobacteria*, *P. aeruginosa,* and *Burkholderia* in the lungs [25,48,138,140,141,142], and *S24_7* and *Odoribacteraceae* families in the gut [139]. Furthermore, the abundance of *Gammaproteobacteria* and *Moraxella* prior to infection showed a strong association to RSV susceptibility [48], but this was contradicted by another study where *Streptococcus* or *Moraxella* infection did not predispose children to RSV infection [141]. Interestingly, this airway-microbiota-associated RSV susceptibility is not seen in adults [50], highlighting the importance of studying the impact of environmental factors on RSV infection in early life. In summary, these studies demonstrate a role for respiratory and gut microbiota in influencing RSV severity and susceptibility to infection in infants (Figure 3).

Studies have also aimed to measure how an individual’s genetic background using genome-wide association studies (GWAS). Single nucleotide polymorphisms (SNPs) have been identified as playing a role in RSV susceptibility or influencing the severity of disease [16,17]. For example, SNPs within the CXCR31 intron have been associated with RSV bronchiolitis [17], but in this particular study, none of the results for detected SNPs reached statistical significance on a genome-wide level. While significant associations between SNPs and RSV have not been observed, these GWAS studies have enabled smaller, hypothesis-driven investigations of candidate SNPs. The literature search for this review yielded some targeted studies of certain SNPs, particularly greater frequencies of IL-10 polymorphisms in RSV and bronchiolitis cohorts compared to control groups of both healthy infants and infants with less severe disease. There is therefore genetic evidence that points to a role of IL10 in RSV pathogenesis. SNPs in the promoter region of the IL-10 gene were investigated in healthy infants and those with RSV, as well as comparing moderate (outpatients) and severe (PICU-admitted) disease cohorts. SNPs were measured in IL-10, and four SNPs were associated with the regulation of its translation (rs1800871, rs1800872, rs1800890, and rs1800896) [143]. In cases of moderate RSV, a variant of rs1800890, which is usually associated with increased levels of IL-10, was less frequent than in controls. This suggested a negative correlation between IL-10 concentration and bronchiolitis severity, but overall associations between IL-10 protein levels and SNPs were not reported [144]. SNPs in two other gene candidates, antioxidant enzymes and NF-E2-related factor 2, were correlated with the requirement for oxygen supplementation and intensive care support in infants with RSV-induced bronchiolitis [145]. Further studies of nasal cytokines and genetic regulators of these cytokines in infants infected with other viruses may resolve the role of these factors in bronchiolitis.

Immunity and inflammatory pathways can also be regulated by miRNAs. The regulation of gene expression by miRNAs is an important process and provides insight into the slight modifications in immune response that can contribute to varying disease outcomes. miRNA expression has been widely implicated in viral infection, as recently reviewed [146]. In a study measuring overall changes in miRNA expression in response to RSV-infection in human alveolar A549 cells, let-7f expression was most abundantly expressed following infection [147]. This expression was partially mediated by the RSV G protein, showing a link between RSV and the host epithelial miRNA. More recently, however, miRNA studies have refocussed towards clinical RSV disease. When investigating the infant antiviral response, studies have identified specific miRNAs differentially expressed in RSV-infection compared to controls, as well as in mild/moderate vs. severe disease. miRNAs measured in the nasal brushings and NPAs of infants less than 12 months of age revealed a reduced expression of miR-34c, miR-34b, and miR-125b in RSV-infected patients compared to healthy, age-matched controls. To take this further, a significant downregulation of miR-125b was measured specifically in mild/moderate disease compared to those with severe symptoms [148,149]. Multiple miRNAs associated with the NF-κB pathway, including miR-125b, were increased in severe disease [149], and although this is suggestive of a reduced antiviral inflammatory response, it would be critical to understand which immune cells are affected by deregulated signalling through this pathway. There were also significant differences in miRNA expression in each of the M2 subtypes, M2a-d [150], where reduced expression of miR-125b in mild/moderate RSV disease could cause a reduced macrophage activation [148] and protect against inflammatory immunopathology. An increased detection of nasal miR-155 levels in infants with respiratory viral infections, including RV, adenovirus, and influenza, was matched with an increased Type-1 response and lower clinical severity scores [151]. Epigenetic regulation of the immune system and the subsequent effects on the response to RSV infection have been reviewed in greater detail elsewhere [152].

## 7. Conclusions

Despite differences in the cell types and cytokines measured in each RSV-infected cohort, it is apparent that Type-2 immunity is associated with RSV disease. This association is particularly clear in the first few months of life, resulting in uncertainty over whether RSV triggers a Type-2 response, or whether a Type-2-biased environment is favoured in the early-life airway and facilitates RSV infection. Immune responses during RSV infection are partly mediated by resident and infiltrating immune cells, but stromal and parenchymal mucosal cells also contribute through mucin production and epigenetic changes evident via miRNAs. Mucin responses to RSV, measured in respiratory samples, offers a possible marker of predisposition to infection in the form of altered mucus composition and decreased mucociliary clearance, which may in turn be affected by the developing respiratory microbiome. Mucins also shape the interactions between immune and epithelial cells, and therefore influence the downstream effects both before and during RSV infection. Although prospective studies show some evidence of differences in immunity in the infants that later become infected by RSV, looking at overall cellular compositions in the respiratory mucosa prior to infection would enable a more complete understanding of the factors governing susceptibility to infection, as well as those influencing disease severities. Human-infection challenge studies in adults may assist in this goal, with recent evidence showing that mucosal neutrophil infiltration predisposes RSV infection [50]. However, studies in adults may not reflect the situation in RSV-naive infants. Another factor that needs to be considered is airway colonisation by commensal bacteria soon after birth. The abundance of specific bacteria in the airway or gut prior to RSV infection may increase susceptibility to disease and contribute to both short- and longer-term clinical outcomes, with there being evidence for specific bacterial phyla abundance in more severe RSV infection. This highlights the importance of including analysis of the lung microbiome, fungome, and virome in future studies, especially when they have the potential to impact the immunological environment of the airways.

## Figures and Tables

**Figure 1 cells-11-01153-f001:**
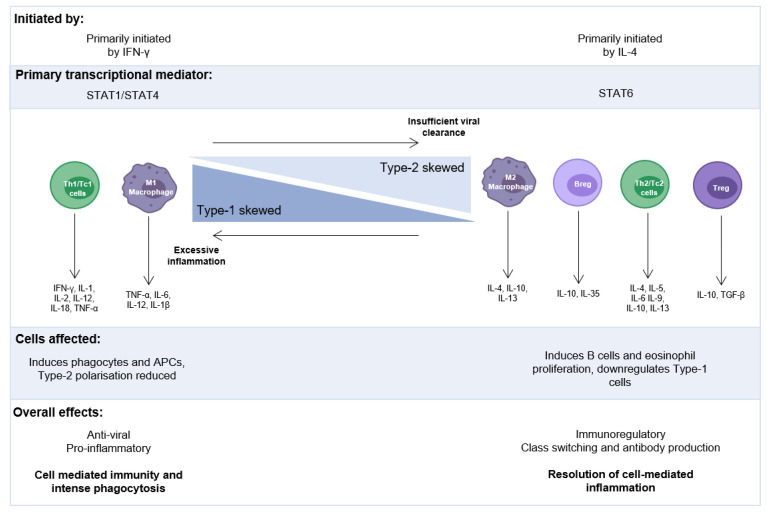
The balance of proinflammatory and immunoregulatory immune responses: Biased, excessive Type-1 or Type-2 responses can have damaging effects to the airways through uncontrolled inflammation or too much protection/reduced viral clearance, respectively. Type-2 skewing has been measured in response to RSV infection in infants characterised by an age-dependent increased concentration in IL-4 and IL-13. Breg: regulatory B cell, IFN: interferon, IL: interleukin, TGF: transforming growth factor, TNF: tumour necrosis.

**Figure 2 cells-11-01153-f002:**
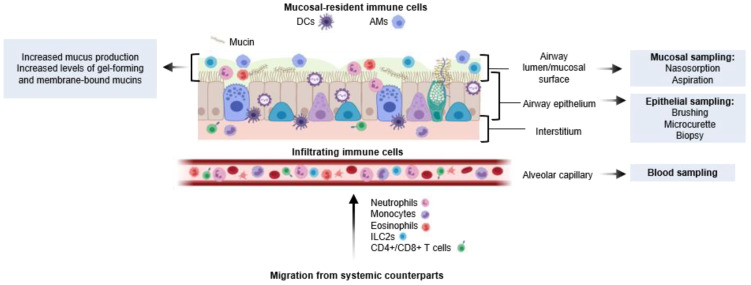
Anatomical distribution of mucosal responses to RSV infection: The respiratory mucosa is colonised by bacteria during and immediately after birth. Resident immune cells (primarily of AMφs and cDCs) respond to environmental factors including commensals, pathogens, and pollutants; they signal to other immune cells that arrive from the circulation. These infiltrating cells move through the mucosa in later stages of infection. AMφs: alveolar macrophages, DCs: dendritic cells, ILC2s: Type-2 innate lymphoid cells, NPA: nasopharyngeal aspirate.

**Figure 3 cells-11-01153-f003:**
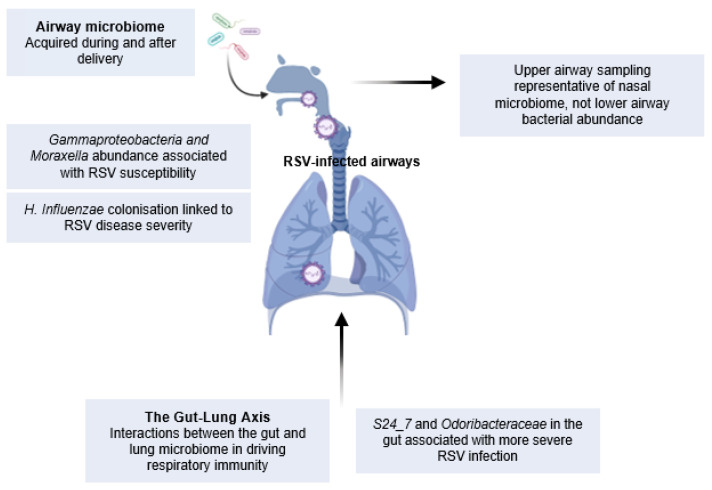
The gut and lung microbiome in RSV infection: The microbiota of the lungs and the gut have the potential to shape respiratory immunity. Distinct bacterial species have been identified in both the lung and gut of RSV-infected infants, and longitudinal sampling pre- and post-RSV infection has revealed that specific bacterial colonisation is linked to susceptibility to infection and can shape longer-term respiratory health.

## Data Availability

Not applicable.

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
