# Peer review of "Mucosal Immune Responses to Respiratory Syncytial Virus"

_cells, 2022, doi:10.3390/cells11071153_

Round 1

Reviewer 1 Report

The authors made some changes to the manuscript. However, the article still lacks easy readability and is confusing to the readers looking for a mucosal immune response to RSV.

As I mentioned before, the title must be following the information of the article. This review mentions the findings in blood against RSV. This point must be incorporated in the title. Accordingly, if I am a reader interested in the mucosal immune response, I must read much unnecessary information found in a basic immune response review o in a book.

In the text, there still are sentences without the proper citations.

  1. Lines 190-196: not correctly cited. (Example)
  2. The conclusions are based on mucosal immune response but not on blood. That is most of the article.

Author Response

Reviewer 1:

The authors wish to thank Reviewer 1 for their considered review of this manuscript. We have made many substantial alterations to the manuscript in light of these comments. In particular we have fundamentally changed the structure of the manuscript such that there is no longer a specific session on observations made in blood. Instead, the results from this section have been substantially abbreviated (to nearly half the word count) and these data have been integrated into the relevant mucosal responses sections. In these instances, we have sought to justify the inclusion of data from blood-based measurements by highlighting the difficulty in performing these studies in the mucosa (particularly given our desire to predominantly highlight human studies). We therefore believe that those blood-based data now included in the manuscript are minimal and essential for supporting the mucosal focus of this manuscript.

We also carefully considered Reviewer 1’s comment on the suitability of the title of the manuscript. Having now enhanced the focus of the manuscript to the respiratory mucosa we believe that this title reflects the main aim and import of the manuscript and have chosen to retain the original title. We would not wish for readers to think that the manuscript comprehensively reviews blood-based changes during RSV infection.

We have responded to Reviewer 1’s specific comments below, including quotes of amended text where relevant. 

Overall comment:

“The authors made some changes to the manuscript. However, the article still lacks easy readability and is confusing to the readers looking for a mucosal immune response to RSV.

As I mentioned before, the title must be following the information of the article. This review mentions the findings in blood against RSV. This point must be incorporated into the title. Accordingly, if I am a reader interested in the mucosal immune response, I must read much unnecessary information found in a basic immune response review or in a book”.

Our response:

We appreciate this feedback, and believe that the aforementioned new structure, and removal of the paragraph detailing blood cytokine responses, helps with this problem. Regarding the basic immunology found in the ‘Background’ section we agree that this can be better be found elsewhere but consider this essential for a more lay audience to interpret the detailed immunological data included in the manuscript. Furthermore, in some instances we have modernised nomenclature e.g. using ‘Type-1’ to include ‘Th1’ and ‘ILC1’ derived immune responses. We believe that this brief introduction to the immunological background is therefore important for easy interpretation of the Type-1/2 biasing hypotheses and other immunological principals later in the manuscript.

Individual comments/suggestions:

In the text, there are still sentences without proper citations.

  1. Lines 190-196: not correctly cited. (Example)
  2. The conclusions are based on mucosal immune responses but not on blood. That is most of the article.

Our response:

  1. The citations have been updated throughout the manuscript and we have sought to ensure that they are included with every study we mention. We have now reiterated references in each line that refers to any given paper, rather than having multiple sentences referring to any single citation. The lines you refer to, now 244-253, have citations for all three studies mentioned in that section:

This continuous transfer of Tregs to the fetus may explain why neonates born pre-term have significantly lower frequencies of CD4+CD25hiFoxP3+ Tregs than term neonates67. As Tregs function to maintain homeostasis and downregulate T cell proliferation68, reduced concentrations could result in exaggerated and damaging inflammatory immune responses to infection in those born prematurely. The importance of Tregs in RSV disease has also been shown in murine models, where abrogating Tregs prior to infection led to greater disease severity, enhanced CD4/CD8 T cell responses and lung eosinophilia compared to WT mice69. The persistence of Th2-type responses with eosinophilia, and significant weight loss, was measured at the beginning of day six and seven post-infection, respectively69. This therefore highlighted the roles of Tregs in preventing this longer-lived pathological Type-2 response to RSV infection69.

  1. The conclusion (now lines 761 to 784) now refers to the mucosal Type-2 response detailed in both the cellular infiltration section, and mucosal cytokine section. We also mention the nasal microbiota and its role in RSV, to try and direct research to consider the role of this in future analyses, where possible:

Conclusion:

Despite differences in the cell types and cytokines measured in each RSV-infected cohort, it is apparent that Type-2 immunity is associated with RSV disease. This association is particularly clear in the first few months of life, resulting in uncertainty over whether RSV triggers a Type-2 response, or if a Type-2 biased environment is favoured in the early life airway and facilitates RSV infection. Immune responses during RSV infection are partly mediated by resident and infiltrating immune cells, but stromal and parenchymal mucosal cells also contribute through mucin production and epigenetic changes evident via miRNAs. Mucin responses to RSV, measured in respiratory samples, offers a possible marker of predisposition to infection in the form of altered mucus composition and decreased mucociliary clearance which may in turn be affected by the developing respiratory microbiome. Mucins also shape the interactions between immune- and epithelial cells, and therefore influence the downstream effects both before and during RSV infection. Although prospective studies show some evidence of differences in immunity in the infants that later become infected by RSV, looking at overall cellular compositions in the respiratory mucosa prior to infection would enable a more complete understanding of the factors governing susceptibility to infection, as well as those influencing disease severities. Human infection challenge studies in adults may assist in this goad, recent evidence showing that mucosal neutrophil infiltration predisposes RSV infection50. However, studies in adults may not reflect the situation in RSV-naive infants. Another factor that needs to be considered is airway colonisation by commensal bacteria soon after birth. The abundance of specific bacteria in the airway or gut prior to RSV infection may increase susceptibility to disease and contributes to both short- and longer-term clinical outcomes, with there being evidence for specific bacterial phyla abundance in more severe RSV infection. This highlights the importance in including analysis of the lung microbiome, fungome and virome in future studies, especially when they can have such an impact on the immunological environment of the airways.

Reviewer 2 Report

No further comments/suggestions

Author Response

Reviewer 2

The authors wish to thank Reviewer 2 for their endorsement of this manuscript. In response to their comment on the English language used in the manuscript we have thoroughly proof-read the manuscript and made numerous grammatical and stylistic corrections to the content that we hope have improved readability. We have also improved the organisation of the manuscript in response to other reviewers comments, which we hope will also improve the language style of the manuscript.

Reviewer 3 Report

This manuscript describes a very powerful and interesting review of the immune response against RSV. This highly experienced group has published other similar studies and the authors are experts in this topic. The work advances the field because it is summarizing the immune response against this virus, aiming in particular to the mucosal response against RSV, which is to date understudied.

There are some weaknesses, listed below. The major concerns are a lack of a clear definition of the outcomes of the review, lack of a detailed methodology description in conducting the review, an organization of the manuscript, and a writing style that may be too complex using large paragraphs which hinders the comprehension and reading of the article.

Background

In my opinion, the Background section would improve if the following is considered:

  • The length should be reduced.
  • The content should be directed to what is known to date and what is the gap in the knowledge the review is aiming to address.
  • The last paragraph in line 108 states different topics being discussed in the review. However, the presentation of the information is confusing. I suggest reviewing this paragraph and focusing to state the true objectives and outcomes of the article.

Methods: The authors present important characteristics in order to understand the research methodology used in the article. However, I suggest adding the following information in order to be able to critically analyze the validity of the article:

  • Which keywords were used for the article search?
  • According to the authors, articles were picked based on evidence. How was this performed? Did they use a scale? Which one?
  • What type of review is it? Literature review, scope review?
  • Which are the main and secondary outcomes of the review?
  • Which were the inclusion and exclusion criteria for the selection of the articles?

Results:

The manuscript would strengthen if the authors include initially in the results section, how many articles were found, how many screened, how many were finally included, and why.

Structure and writing of the manuscript:

The writing of the manuscript focuses on the general response to RSV and also the mucosal response. I suggest focusing only on the latter, which seems to be the true objective of the article mentioned in the title of the work. This is associated with a lack of a clear definition of the objectives of the study previously mentioned.

For example, there are two sections to discuss the role of cytokines in the response to RSV (4 and 6). This prevents fluid and organized reading of the results.

Conclusion:

The first paragraph summarizes some of the findings of the review. However, the following paragraphs mention new topics not previously discussed. I suggest focusing the conclusions on the findings of the article.

In general, the authors carry out an extensive and very interesting analysis and review of the topic, however, I consider that the concerns mentioned above would substantially improve the article.

Author Response

Reviewer 3:

Overall comment:

This manuscript describes a very powerful and interesting review of the immune response against RSV. This highly experienced group has published other similar studies and the authors are experts in this topic. The work advances the field because it is summarizing the immune response against this virus, aiming in particular to the mucosal response against RSV, which is to date understudied.

There are some weaknesses, listed below. The major concerns are a lack of clear definition of the outcomes of the review, lack of detailed methodology description in conducting the review, an organization of the manuscript, and a writing style that may be too complex using large paragraphs which hinders the comprehension and reading of the article.

Our response:

Thank you for your detailed feedback regarding the manuscript, in particular what we can do to improve the quality of writing. We have incorporated the primary aim of the review, which we will detail below, provided more detail about the generation of relevant studies in the methodology and results, and have changed the structure substantially. We believe that these amendments have significantly improved the manuscript and enhanced the mucosal focus of the text.

Individual comments/suggestions:

Background:

In my opinion, the Background section would improve if the following is considered:

The length should be reduced.

The content should be directed to what it known to date and what is the gap in the knowledge the review is aiming to address.

The last paragraph in line 108 states different topics being discussed in the review. However, the presentation of the information is confusing. I suggest reviewing this paragraph and focusing to state the true objectives and outcomes of the article.

Our response:

We agree, and have reduced the length of the background section, particularly the section regarding the Type-1/Type-2 immunology. We have better addressed the gap in the knowledge and followed this with our primary aims for the review, as shown below (Lines 87-108):

However, we are only just starting to understand how immune- and epithelial cells communicate at the first interaction with a virus, as the balance between an efficient and damaging local inflammatory response can become dysregulated, especially in early life.

The primary aim of this review is to collate results on the mucosal response to RSV infection, including both immune- and epithelial cell responses. This includes the production of cytokines in response to infection, but also the production of mucins, the major structural component of the mucus lining the epithelial surfaces in the airways and gastrointestinal tract, with their concentrations determining the viscosity of the mucus itself. We also investigate the evidence for immunoregulation in early life, characterised by signalling by immunoregulatory immune cells and Type-2 skewing. In addition, the discussion will focus on whether there are correlations between the Type-1 and Type-2 responses to RSV in the mucosa, with a focus on disease severity, and finally how it compares to responses seen in blood, and to other respiratory viruses. We highlight the importance of sampling the mucosa when studying RSV disease severity and exacerbation, the role of cellular immune responses, and immunosuppressive mechanisms.

Methods:

The authors present important characteristics in order to understand the research methodology used in the article. However, I suggest adding the following information in order to be able to critically analyse the validity of the article:

Which keywords were used for the article search?

According to the authors, articles were picked based on evidence. How was this performed? Did they use a scale? Which one?

What type of review is it? Literature review, scope review?

Which are the main and secondary outcomes of the review?

Which were the inclusion and exclusion criteria for the selection of the articles?

Our response:

Thank you for this feedback, we agree that including this will improve the robustness of the review, so have incorporated these suggestions into the writing. We initially designed this manuscript as a systematic literature review to select articles from a broad literature search in PubMed. It was only after we had obtained our list of relevant studies, we performed additional analysis on studies performed in cell culture and mouse models, which is one of the factors preventing this from being a scoping review. These cell culture and mouse model papers were included to highlight commonalities and important disparities between study approaches. We have included far greater detail on this design in the Methods section of the manuscript.

Please see a copy of the new Methods section, lines 110-133 in the manuscript:

Methods:

To generate the studies for this literature review, an initial systematic approach was employed, whereby an advanced literature search was performed through PubMed, with the final search being conducted on 27th October 2020 using the following search terms:

((infan*) OR (child*) OR (pediat*) OR (paediat*)) AND ((lung*) OR (airway*) OR (upper respiratory tract) OR (lower respiratory tract) OR (respiratory tract*)) AND (infant) AND ((RSV) OR (respiratory syncytial virus))

These search terms were primarily used to capture all studies investigating immune responses to RSV in those under the age of two, whether this was measured in blood or the airway mucosa. Specific topics were followed up with additional literature, where the original search terms did not capture studies known to the authors or found in paper references. For example, three studies25,47,48 were added into Section 5 as these studies were published after the date of the literature search, but supported the data originally collated from the relevant studies. There was a specific focus on human infant studies but results from mouse models and in vitro studies were also included to support the results from human in vivo studies. This review defined “neonates” as those up to 28 days, and “infants” included children aged up to two years, though the definition of “infant” by age varied throughout the studies. Results were therefore included if the age range of participants in a study did not surpass two years of age. Once reviews and studies not relevant to this specific review were filtered out, the results were collated and organised by specific measurements, sample type, age-related differences in responses and the response to RSV compared to other viruses. While the immune response of the respiratory mucosa was the primary remit of this review, studies of peripheral blood measurements were included where these complemented studies of the mucosa or filled knowledge gaps on mucosal immunity to RSV.

Results:

The manuscript would strengthen if the authors include initially in the results section, how many articles were found, how many were screened, how many were finally included, and why.

Our response:

We agree, and this follows on nicely from the detail in the Methods section. We have therefore included an initial paragraph providing these details (Lines 136-147):

After using the search criteria specified in the Methods, the literature search returned 4,572 studies which were then screened based on their title and abstracts if they: measured immune cell populations, cytokine and mucin concentrations, or RNA expression in the blood and/or upper respiratory mucosa during RSV infection. After excluding epidemiology or surveillance-based studies, those just using animal models, or investigations into vaccine candidates and testing, there were 79 studies that were relevant to the review. As also described in the Methodology, some topics were followed up independently if the literature search did not capture all relevant studies, or if there was less evidence in human studies. Results from cell culture or mouse models of RSV infection were included in the sections pertaining to immunoregulatory cells during infection, as this has been less well-studied in infants. Furthermore, these important cell culture and mouse model studies can be evaluated to help direct future research conducted in vivo.

Structure and writing of the manuscript:

The writing of the manuscript focuses on the general response to RSV and also the mucosal response. I suggest focusing only on the latter, which seems to be the true objectives of the article mentioned in the title of the work. This is associated with a lack of a clear definition of the objectives of the study previously mentioned.

For example, there are two sections to discuss the role of cytokines in the response to RSV (4 and 6). This prevents fluid and organised reading of the results.

Our response:

We thank the reviewer for this prescient comment. We have taken this into consideration and agree that the structure needed improvement to aid the readability. We have incorporated the two cytokine paragraphs, only briefly mentioning the blood results as a comparison to what we see in the mucosa (Section 3.1, lines 456-458), please see below.

This agrees with what has been measured in the plasma/serum of infants with PCR-confirmed RSV infection, where significant increases in IL-6, IL-10, IL-12, IL-13 and IL-35 levels have been reported compared to controls, indicative of a robust Type-2 response32,39,93,100.

Furthermore, thanks to your suggestions, we believe that the new structure of the manuscript improves fluidity and helps with the focus on the mucosa, instead of blood and mucosal responses. This is manifest in a considerable abbreviation of the blood-based data that is included in the manuscript and incorporation of these sections into the relevant mucosal immunity sections. These changes mean that the blood data is brief and incorporated into the rest of the text, rather than having stand alone sections. We believe that this has helped to retain the mucosal focus of the manuscript, which the Reviewer correctly highlights as our main aim.

Conclusion:

The first paragraph summarizes some of the findings of the review. However, the following paragraphs mention new topics not previously discussed. I suggest focusing the conclusions on the findings of the article.

In general, the authors carry out an extensive and very interesting analysis and review of the topic, however, I would consider that the concerns mentioned above would substantially improve the article.

Our response:

Thank you for your feedback, and for taking the time to suggest these important improvements. The conclusions have been extensively re-written to ensure we only discuss what has been evaluated in the review, as well as trying to direct future research (Lines 761-784):

Conclusion:

Despite differences in the cell types and cytokines measured in each RSV-infected cohort, it is apparent that Type-2 immunity is associated with RSV disease. This association is particularly clear in the first few months of life, resulting in uncertainty over whether RSV triggers a Type-2 response, or if a Type-2 biased environment is favoured in the early life airway and facilitates RSV infection. Immune responses during RSV infection are partly mediated by resident and infiltrating immune cells, but stromal and parenchymal mucosal cells also contribute through mucin production and epigenetic changes evident via miRNAs. Mucin responses to RSV, measured in respiratory samples, offers a possible marker of predisposition to infection in the form of altered mucus composition and decreased mucociliary clearance which may in turn be affected by the developing respiratory microbiome. Mucins also shape the interactions between immune- and epithelial cells, and therefore influence the downstream effects both before and during RSV infection. Although prospective studies show some evidence of differences in immunity in the infants that later become infected by RSV, looking at overall cellular compositions in the respiratory mucosa prior to infection would enable a more complete understanding of the factors governing susceptibility to infection, as well as those influencing disease severities. Human infection challenge studies in adults may assist in this goad, recent evidence showing that mucosal neutrophil infiltration predisposes RSV infection50. However, studies in adults may not reflect the situation in RSV-naive infants. Another factor that needs to be considered is airway colonisation by commensal bacteria soon after birth. The abundance of specific bacteria in the airway or gut prior to RSV infection may increase susceptibility to disease and contributes to both short- and longer-term clinical outcomes, with there being evidence for specific bacterial phyla abundance in more severe RSV infection. This highlights the importance in including analysis of the lung microbiome, fungome and virome in future studies, especially when they have the potential to impact the immunological environment of the airways.

Round 2

Reviewer 3 Report

All my comments were addressed, and substantial changes were made to the article.

Author Response

We thank the reviewer for this endorsement and for their efforts in helping us to improve this manuscript.